# Virtual Screening and Quantitative Structure–Activity Relationship of *Moringa oleifera* with Melanoma Antigen A (MAGE-A) Genes against the Therapeutics of Non-Small Cell Lung Cancers (NSCLCs)

**DOI:** 10.3390/cancers14205052

**Published:** 2022-10-15

**Authors:** Smitha S. Bhat, Shreya Das Mahapatra, Sindhu R, Sarana Rose Sommano, Shashanka K. Prasad

**Affiliations:** 1Department of Biotechnology and Bioinformatics, JSS Academy of Higher Education and Research, Mysuru 570 015, Karnataka, India; 2Department of Microbiology, JSS Academy of Higher Education and Research, Mysuru 570 015, Karnataka, India; 3Plant Bioactive Compound Laboratory, Faculty of Agriculture, Chiang Mai University, Chiang Mai 50100, Thailand; 4Department of Plant and Soil Sciences, Faculty of Agriculture, Chiang Mai University, Chiang Mai 50200, Thailand

**Keywords:** lung cancer, non-small cell lung cancer, NSCLC, *Moringa oleifera*, phytocompounds

## Abstract

**Simple Summary:**

A variety of phytochemicals found in *Moringa oleifera* have been found to be biomedically active. MAGE-A variants are expressed in most carcinoma cells, and the current study aims at the computational discovery of phytochemicals in treating NSCLC via MAGE-A targeting.

**Abstract:**

In the last decade, there have been significant advancements in the treatment of non-small cell lung cancer, including remarkable gains in detection, diagnosis, and therapy. The emergence of molecular targeted therapies, immunotherapeutic inhibitors, and antiangiogenesis medicines has largely fueled improvements in combination therapy and systemic treatments, all of which have dramatically ameliorated patient outcomes. The *Moringa oleifera* bioactive compounds have been effective in the suppression of cancers, making them the therapeutic agents of choice for the current investigation to treat MAGE-A presented in NSCLC. The ligand entrants were screened for their pharmacological properties, and 2,2-diphenyl-1,3-benzodioxole was stipulated as the lead candidate. 2,2-Diphenyl-1,3-benzodioxole exhibited better pharmacological properties and superior binding with branched-chain amino acids, making it an ideal candidate to address MAGE-A. The study concluded that addressing MAGE-A to impede their activity and antigenicity can be exploited as immunotarget(s).

## 1. Introduction

Despite recent therapeutic improvements, NSCLC (non-small cell lung cancer) is still a serious concern in terms of public well-being, with more cancer-related deaths than any other malignancy in both males and females [1]. Pulmonary melanoma’s clinical manifestations can be attributed to bronchial and alveoli lining destruction. The two kinds of lung cancer are small cell lung cancer, and NSCLC, which account for 15 and 85 percent of total lung cancer cases, respectively. Adenocarcinoma, squamous, and giant cell carcinomas are the three major categories of NSCLC. NSCLC can harm any form of lung cell, including alveolar, bronchial, epithelial, squamous, and other pulmonary cells [2,3,4,5].

Adenocarcinoma is frequently encountered among women and is localized to the exterior portion of the pulmonary tissues and normally destroys the mucosal cells. It is also known as bronchoalveolar carcinoma, which can affect both smokers and non-smokers. Squamous cell cancer occurs in the squamous cells that border the esophageal and bronchial tree and is concentrated in the center of the lungs [6]. Large cell carcinoma is undifferentiated cancer that can arise in any area of the lung and has the potential to grow and spread quickly. NSCLCs are more common among the elderly, with the average age of the patients being 70 years. These cancers are usually metastatic at presentation and need immediate management [6,7]. Since the elderly are the target population, it makes therapies challenging. In recent years, various immunotherapy methods have exposed immune manipulation strategies to mediate the regression of malignancies and cure entails the immunization process in contrast to CD 4 and 8+ T cells to eradicate the antigen-expressing cancer cells [8,9]. Melanoma-associated gene expression antigens (MAGE-A) have been identified in melanoma cell lines, and the Xq28 chromosome encodes the entire protein family of this gene. The articulation of MAGE-A genes has been documented in a multitude of diseases, and NSCLC expresses a variety of MAGE-A genes on a systematic basis [10]. The MAGE-A clusters, which encompass 12 subgroups extending from A1 to A12, have been identified most consistently in malignant tumors. MAGE-A variants are expressed in different carcinoma cells; nonetheless, the prevalence of MAGE-A expression varies between tumor types and subtypes, while MAGE-A translation is recognized to be modulated by epigenetic mechanisms. However, the underlying factors that contribute to their variation remain unexplained [11]. Numerous studies have suggested that certain MAGE-A proteins may correspond with a poor therapeutic response(s). MAGE-A proteins remain immunogenic and are key targets for new therapeutic strategies in NSCLC, though their physiological role in cancer pathogenesis remains undiscovered [12]. All eukaryotes share MAGE genes, which have quickly increased in number in mammals. Based on tissue expression patterns, the human MAGE family can be classified into two groups: type I MAGEs consisting of MAGE-A, -B, and -C and type II MAGEs (MAGE-D, -E, -F, -G, -H, -L, and Necdin) [13,14]. Type I MAGEs are considered cancer–testis antigens, and type II are expressed throughout various tissues in the body. A MAGE homology domain with a length of about 170 amino acids exists in both type I and type II MAGEs, and it is typically 46% conserved across all human MAGEs [15]. In NSCLC tumors, while MAGE-A3 and -A9 expression levels have been substantially linked with shorter survival times, few studies have looked at MAGE-A3 expression as a probable independent pointer of poor prognosis in NSCLC patients [16,17,18]. The recombinant MAGE-A3 protein was therapeutically administered in nearly 3,000 patients following the resection of lung cancer tumor in the largest therapeutic study for lung cancer, MAGRIT (MAGE-A3 as Adjuvant NSCLC Immunotherapy) [19]. MAGE-A4ᶜ¹º³²T is currently undergoing phase I clinical testing to assess its safety and tolerability for a number of malignancies. [20].

Natural ingredient-based solutions have been established to be useful in the treatment and prevention of an assortment of diseases and conditions. Plants have been utilized in folk treatment owing to their therapeutic properties, and different plant parts, such as foliage, flowers, branches, roots, stems, and bark, are maneuvered into discrete forms to be implemented as drugs. Herbal medicines promise better lifestyle management with increased health benefits, and reduced risks and side effects [21]. Based on the literature, we considered *Moringa oleifera* (*M. oleifera)* phytochemicals as therapeutic entrants to address MAGE-A expressed in NSCLC [22,23,24]. *M. oleifera* extracts have been demonstrated to possess promising antineoplastic activity against a variety of cancers. The *in vitro* cytotoxic tests of the leaf and bark extracts revealed the arrest of the cell cycle at the G2/M phase in both breast and colorectal cancers [25]. Furthermore, water extracts were found to bring about intensification in the reactive oxygen species, leading to p53 activation and breakage of the poly(ADP–ribose) polymerase 1 enzyme, eventually causing apoptosis in lung cancer cells [26]. Aqueous extracts have also induced time- and concentration-dependent cytotoxicity in Ehrlich ascites carcinoma and laryngeal carcinoma by modifying the mitochondrial membrane potential, prompting apoptosis [27]. Furthermore, these extracts exhibited a dose-dependent inhibition of cell proliferation, which was evidenced by in vivo studies in the liver- and lung-cancer-bearing mouse models [28]. Similar results were reported by other researchers using rat models [29]. Although the extracts of *M. oleifera* have favored pro-apoptosis events in many cancer cell lines, studies highlighting molecular targets for therapeutic purposes are less frequently reported. Identifying effective therapeutic targets plays a vital part in the treatment of any disorder including cancer. Only a handful of computational analyses reported on the antitumor potential of *M. oleifera* bioactive compounds. The phytochemicals of *M. oleifera* were evaluated against Bcl-2-associated X protein using a computational approach, revealing the successful docking of ligands to the protein [30]. The studies by Apeh et al. [31] successfully identified β-Sitosterol as a potent anticancer agent against breast and prostate cancers, with strong binding capabilities against human placenta aromatase, poly(ADP–ribose) polymerase, phosphoinositide-3-kinase, tumor necrosis factor-α, and caspase 8 proteins [31]. These studies revealed the efficacy of computational tools in identifying the potential molecular targets in cancer and their interaction with phytobioactives. The current work focuses on computational evaluation via virtual screening and molecular docking simulations of *M. oleifera* bioactive compounds against MAGE-A, a viable treatment modality for NSCLC management.

## 2. Materials and Methods

An incremental and exhaustive virtual screening and molecular docking analysis were performed to uncover the most promising therapeutic targets in pursuit of a lead that might impede NSCLC serotypes.

### 2.1. Compilation of the Dataset and Ligand Preparation

The 3D structures of the phytoconstituents from *Moringa oleifera* were retrieved from the PubChem database (https://pubchem.ncbi.nlm.nih.gov/(accessed on 13 June 2022)) [32]. The SDF format (standard data format) was used to extract all the compounds from the default PubChem database. Using OpenBabel (v2.4.1) (https://sourceforge.net/projects/openbabel/files/openbabel/2.4.1/ (accessed on 18 June 2022)), the .sdf files were converted to .pdb format and was purified using the Biovia Discovery visualization tool (v21.1.0.20298, Dassault Systèmes) (https://discover.3ds.com/discovery-studio-visualizer-download (accessed on 18 June 2022)) [33,34].

### 2.2. Homology Modeling of MAGE-A

Comparative modeling is the only approach that can accurately construct a three-dimensional model for a protein from its presented amino acid composition, as opposed to all other methodologies. MAGE’s 3D model topology was generated, projected, and evaluated using the Swiss Model (https://swissmodel.expasy.org/ (accessed on 13 June 2022)) [35].

### 2.3. Refinement of Validation of the Model

The projected model went through an energy reduction procedure implemented by using DS BIOVIA Discovery Studio and evaluated by using Swiss PDB Viewer (v4.1.1) (https://spdbv.unil.ch/ (accessed on 13 June 2022)) [36]. The assessment of the modeled structure was accomplished using a structural assessment and evaluation server. A crucial aspect of the comparison modeling procedure is the evaluated output of the modeled proteins from the server.

### 2.4. Prediction of Active Binding Site

To determine the active binding site, CASTp (http://sts.bioe.uic.edu/castp/calculation.html (accessed on 18 June 2022)) was utilized to extract all potential binding or predicted sites. Such chains could be found on the surface and as well as inside proteins. CASTp, a visual interface software, displays the user-uploaded structures’ flexible interactive representation as well as the calculation [37].

### 2.5. In Silico Preclinical Testing of Phytocompounds

The process of drug discovery, research, and development is marked by a high level of complexity and is related to technical advancements to track its progress. Several biotechnological tools associated with medicinal chemistry approaches have disclosed a prominent role in the development process of novel molecules with biological activity.

The computer-assisted screening technique minimizes the likelihood of failure while also saving time and resources. To assess the pharmacological properties, the SwissADME web browser (http://www.swissadme.ch (accessed on 18 June 2022)) was utilized to evaluate the compounds for ADME, physiochemical, drug-likeness, pharmacokinetics, and medicinal chemistry parameters [38,39].

#### 2.5.1. Drug Likeness, ADMET Analysis, and Prediction of Toxicity

SwissADME and pre-ADMET analyses with respect to the five rules of Lipinski filter analysis were used to examine the absorption, distribution, metabolism, excretion, and toxicity (ADMET), and drug-likeness analyses. There are various established criteria for analyzing orally active drugs, such as cLogP, molecular mass, and hydrogen bond donor and acceptor. SwissADME, a drug discovery tool, was used to examine or select all the physicochemical features of phytochemical constituents [38,39]. The toxicity of the compounds was predicted using the online Protox II suite (http://tox.charite.de/protox_II/ (accessed on 18 June 2022)) ), which categorizes the compounds into six classes based on their toxic doses.

#### 2.5.2. BOILED-Egg Analysis

The BOILED-Egg method is used to forecast drug development based on gastrointestinal absorption and blood–brain barrier permeability. According to the BOILED-Egg plot, adequately positioned compounds in the white part of the egg have the likelihood of greater GI absorption, and the chance of brain barrier permeability is higher for the compound perfectly positioned in the yellow zone. The SwissADME webserver was used to investigate the chosen compounds for BOILED-Egg analysis [38,39].

### 2.6. Quantitative Structure–Activity Relationship Analysis

#### 2.6.1. Collection of Datasets

A library of 14 *M. oleifera* phytocompounds with known pharmacological properties was prepared based on the results documented in previous research (Table 1).

#### 2.6.2. Optimization of the Geometry

The chemical structures of the above phytocompounds (Table 1) were acquired from the PubChem database, and each molecule underwent molecular mechanic preoptimization before being reoptimized with the 3D QSAR tools (https://www.3d-qsar.com/ accessed on 18 June 2022)) [40,41].

#### 2.6.3. Molecular Descriptor Calculation

Molecular descriptors, on the other hand, are mathematical variables that elucidate various properties of the molecules. The descriptor calculation for all phytochemicals was accomplished with the aid of the QSAR build tool (https://www.computabio.com/3d-qsar-service.html?gclid=EAIaIQobChMIm7umucHf-gIVTTErCh0zEAiPEAAYASAAEgLHzvD_BwE (accessed on 18 June 2022)) [40,41,42].

#### 2.6.4. QSAR Model Generation

In terms of model generation, genetic algorithm techniques were employed to produce a QSAR model (multilinear regression model) utilizing the 3D-QSAR build and toolbox [40,41,42].

### 2.7. Virtual Screening

PyRx (V0.8) (https://pyrx.sourceforge.io/ (accessed on 18 June 2022)) was employed for the virtual screening examination of all the identified phytocompounds in this work. The structures were subjected to the removal of all water molecules and heteroatoms. Additionally, before employing the docking tools, the protein was given Gasteiger charges and hydrogen bonds. PyRx was used to perform multiple ligands docking in the AutoDock vina wizard environment, and configurations were assessed using the RMSD (root mean square deviation) values, lowest energy conformer, and hydrogen bond interactions [43].

### 2.8. Molecular Dynamic Simulations Analysis (MD Simulations)

The best-docked complexes with MAGE-A were subjected to MD simulations to confirm the compound that had the highest binding affinity and the most desirable molecular interaction topology. Desmond–Schrodinger (v20.4, Schrödinger, Inc.: New York, NY, USA) was employed for molecular docking analysis as well as for checking the compounds’ stability [44]. The Desmond–Schrodinger explicit solvent MD simulations were performed for 300 ns to analyze the stability of the compounds against MAGE-A, and the OPLS force field was used to produce ligand topology files. Ligand–protein complexes were also solvated in an octahedron box to validate the precise simulated structures.

## 3. Results

In total, 14 phytobioactives of *M. oleifera* were identified and selected based on a literature survey and subjected to 3D-QSAR and molecular docking techniques to evaluate their activity against MAGE-A protein.

### 3.1. Ligand Dataset

The chosen ligand dataset is listed in Table 1.

### 3.2. Homology Modeling

The disparity between predicted protein sequences and empirically determined structures can be bridged by homology modeling. Protein 3D structures deliver significant input into their molecular functioning and reassure their myriad application(s) in scientific research. For a complete understanding of the biological processes, how protein structures and pathways operate, and systemic manipulations, a complete description of protein associations and their general quaternary structure is necessary.

The MAGE-A protein sequence (Gene ID: 4100) was obtained in FASTA format from NCBI and used to find templates in the Swiss Model. A comparison of the templates was carried out to confirm whether the highest-scored templates and alignments indicate different conformation or represent distinct sections of the target molecule. A 3D protein model of the MAGE-A protein (Figure 1) was immediately obtained for each given template, and in the Swiss Model, the QMEAN scoring tool was used to measure modeling inaccuracies and provide predictions on model accuracy.

### 3.3. Prediction of Active Binding Site

Two binding pockets were considered for the present study based on the drug score (Figure 2). A complete description of the binding pocket is tabulated in Table 2.

### 3.4. In Silico Preclinical Testing of Phytocompounds

#### 3.4.1. Drug Likeness, ADMET Analysis, and Prediction of Toxicity

The drug-likeness analysis helps to predict the potency of the drug candidate to become an oral drug based on the bioavailability score. The scoring parameters are fundamentally based on the structural properties of small molecules. The Lipinski rule of five is used to filter the small molecules and to assess their drug likeness (Table 3 and Table 4). The medicinal chemistry properties of the drug molecules are demonstrated via the PAINS score, which demonstrates the substructures that display robust reactions in the experiments regardless of the protein target.

Lipinski filter criteria: The molecule weight of H-bond donors must be between 150 and 500 g/mol, and the molecular weight of H-bond acceptors must be between 5 and 10. The Lipinski filter analysis passed all the phytochemicals (Table 4), indicating them to possess potential medicinal characteristics, as their values were within the permissible range for human use. ADME analysis details are tabulated in Table 5.

All 14 selected ligands were observed to be non-toxic to vital organs (most importantly the liver), non-immunotoxic, and non-mutagenic. From the toxicity prediction analysis (Table 6), cis-vaccenic acid displayed fatal toxicity with 5 < LD_50_ ≤ 50 mg/kg body weight. Hexadecanoic acid, 2,2-diphenyl-1,3-benzodioxole, cyclohexanone, 2-(3-chloro-2-butenyl) -2-methyl-6,6-diphenyl, palmitoyl chloride, 3-chloro-N-isochroman-1-ylmethyl-propionamide, L-galactose, 6-deoxy-, 2-pyrazoline,1-isopropyl-5-methyl, and 4H-pyran-4-one,2,3-dihydro-3,5-dihydroxy-6-methyl were found to belong to class IV toxicity (harmful if swallowed), with those having higher LD_50_ values between 300 and 2000 mg/kg belonging to class V, which may be considered harmful when ingested, and the compounds (Z)-1-(1-ethoxyethoxy)hex-3-ene, phenacylidene diacetate, and 3,7,11,15-tetramethylhexadec-2-en-1-ol/phytol presented with toxic doses between 2000 and 5000 mg/kg body weight. 1,2,3-Cyclopentanetriol, L-galactose, 6-deoxy- and piperazinedione,4-benzoyl-,2-oxime were found to be non-fatal and non-harmful, belonging to class VI, with LD_50_ greater than 5000 mg/kg body weight. Hence, these compounds were considered safer to use when compared with cis-vaccenic acid based on the data.

#### 3.4.2. Boiled-Egg Analysis

The white section (egg white) represents a greater likelihood of passive absorption through the GI tract, while the yellow region (egg yolk) indicates a high potential for brain invasion (Figure 3). The egg-yolk and egg-white sections are not mutually exclusive. The spots are also colored blue if they are projected to be effectively effluxed by P-glycoprotein (PGP)+ and in red if they are anticipated to be non-substrate of PGP-.

### 3.5. QSAR Analysis

#### 3.5.1. Biological Activity

The constructed dataset was run through PyMol-Edit to determine the comparative molecular field analysis (CoMFA) and comparative molecular similarity index analysis (CoMSIA) values, which were then compared with the MLR outcomes. The most extensively utilized methods for generating 3D-QSAR are CoMFA and CoMSIA. The CoMFA and CoMSIA descriptors were created by deploying a superposed chemical 3D lattice structure separation using a grid distance of two.

The steric, hydrogen bond acceptor (HBA), electrostatic, hydrogen bond donor (HBD), and hydrophobic effects were all quantitatively involved in this study. SYBYL was used to perform a 3D-QSAR assessment with the standard settings (Table 7).

#### 3.5.2. Alignment

The reference conformer was then utilized to align the training and test set compounds using modified thresholds and the maximum common substructure (MCS). The conformation quest was carried out using a very accurate and time-consuming calculation method, with a limit of 500 conformations per molecule. The 3D-QSAR model was created using the least active configurations in comparison to the reference. To produce the best possible model, all orientations were meticulously validated. Using the random selection method, the original training dataset of 6 ligands was then separated into the training and test sets.

The maximum number of components used in QSAR modeling was 20, and the maximum distance between sample locations was 1.0 (Figure 4).

#### 3.5.3. C.Y Randomization Test

To assess the model’s resiliency, the Y randomization test was utilized. By randomly mixing the dependent variable after each cycle, a new QSAR model was constructed (-loglC_50_). The created 3D-QSAR model with original data had low q^2^ and r^2^ values, indicating that it is powerful and not inferred by coincidence (Table 8 and Table 9).

#### 3.5.4. Contour Map Analysis

The contour maps for CoMFA/CoMSIA were constructed to discover the structural requirements that influence binding affinity, which could lead to a boost in the molecules’ biological activity (Figure 5).

Figure 6 shows the most active molecule, 2,2-diphenyl-1,3-benzodioxole, as a reference structure across the contour maps of CoMFA and CoMSIA. The steric, electrostatic, H-bonding, and hydrophobic interaction fields were studied using the CoMSIA/CoMFA contour plot technique to reveal the important molecular features.

### 3.6. Molecular Docking and Visualization

Table 10 lists the binding affinity of each selected ligand for MAGE-A, as determined by PyRx, indicating the strength of the interaction between each ligand and MAGE-A.

The docking conformation that yielded the highest binding energy was considered for further evaluation (Figure 7).

### 3.7. Molecular Dynamic Simulation

2,2-Diphenyl-1,3-benzodioxole was selected for molecular dynamic studies against MAGE-A to investigate the root mean square deviation (RMSD) and root mean square fluctuation (RMSF) calculations. We carried out a molecular dynamic simulation using the Desmond–Schrodinger v20.4 unit. We prepared all the protein topologies and ligand correlations (Figure 8 using the OPLS force field, which is an in-built force in Schrodinger. According to the RMSD and RMSF graphs (Figure 9 and Figure 10), we observed a very minimal fluctuation rate in the protein trajectories. For a better outcome, we ran the protein apo simulation to observe the possible mutation or variation in the protein trajectories and residues. Figure 11 depicts the protein secondary structure histogram.

## 4. Discussion

Significant advancements in the detection of non-cell lung cancer were accomplished in the recent decade, including advances in disease processes and malignancy progression routes, as well as accurate diagnosis and specialized care. With a strong belief that molecularly delineated subtypes are therapeutically exploitable, novel molecular diagnostics and tailored treatments for NSCLC are being continuously explored to identify and manage cancer [45].

Several MAGE peptides are only prevalent in reproductive cells; nevertheless, they are abnormally amplified in cancers. MAGEs were first found as antigens on cancerous cells, and later, they were targeted for cancer immunotherapy. Notwithstanding, current research findings reveal that MAGE peptides play significant roles in carcinogenesis [15]. In NSCLC tumors, MAGE-A3 and -A9 activation is linked to a worse probability of survival. MAGEs are also linked to a higher likelihood of tumor reappearance after treatment. Different cancer forms grow accustomed to MAGEs for survival when they are expressed, such as that of MAGE-As or MAGE-Cs in lung cancer, which boost their invasiveness [46,47].

The current study employed 14 phytocompounds (hexadecanoic acid, cis-vaccenic acid, 2,2-diphenyl-1,3-benzodioxole, cyclohexanone,2-(3-chloro-2-butenyl) -2-methyl-6,6-diphenyl, palmitoyl chloride, piperazinedione,4-benzoyl-,2-oxime, 3-chloro-N-isochroman-1-ylmethyl-propionamide, (Z)-1-(1-ethoxyethoxy)hex-3-ene, phenacylidene diacetate, L-galactose, 6-deoxy-, 3,7,11,15-tetramethylhexadec-2-en-1-ol/phytol, 2-pyrazoline,1-isopropyl-5-methyl, 4H-pyran-4-one, 2,3-dihydro-3,5-dihydroxy-6-methyl, and 1,2,3-cyclopentanetriol) from *Moringa oleifera*, which were selected based on their reported bioactivities. All 14 compounds underwent stringent in silico pharmacological analysis and were assessed for various parameters such as physicochemical properties, topological surface area, number of flexible bonds, toxicity, number of hydrogen atom donors and acceptors, molecular weight, etc. From the results of the analyses, it is evident that all the selected compounds had druggable parameters and passed the filters of Lipinski and ADMET analysis. However, these compounds displayed varied GI absorption and BBB permeability. Of the 14 compounds screened in the current study, 10 of them (hexadecanoic acid, cis-vaccenic acid, 2,2-diphenyl-1,3-benzodioxole, piperazinedione,4-benzoyl-,2-oxime, 3-chloro-N-isochroman-1-ylmethyl-propionamide, (Z)-1-(1-ethoxyethoxy)hex-3-ene, phenacylidene diacetate, 2-pyrazoline,1-isopropyl-5-methyl, 4H-pyran-4-one, 2,3-dihydro-3,5-dihydroxy-6-methyl, and 1,2,3-cyclopentanetriol) displayed high GI absorption, while 5 of the 14 compounds (hexadecanoic acid, 2,2-diphenyl-1,3-benzodioxole, 3-chloro-N-isochroman-1-ylmethyl-propionamide, (Z)-1-(1-ethoxyethoxy)hex-3-ene, and phenacylidene diacetate) exhibited BBB permeability [48,49].

All the selected molecules were subjected to 3D-QSAR analysis, and 2,2-diphenyl-1,3-benzodioxole was utilized to develop the contour maps. The ligands were further subjected to molecular docking, and the results of docking corroborated the previous findings from QSAR analysis with 2,2-diphenyl-1,3-benzodioxole being the compound with the highest binding affinity. The binding of 2,2-diphenyl-1,3-benzodioxole with MAGE-A was visualized, and it was noticed that the ligand had established better associations at MET A:201, GLU A:217, MET A:221, ILE A:197, VAL A:200, VAL A:286, VAL A:283, LEU A: 153, etc. As they operate as nitrogen donors, branched-chain amino acids are essential for cancer cell growth and development. The validation of docking results revealed that 2,2-diphenyl-1,3-benzodioxole established bonds with branched-chain amino acids. Although not much biomedical activity of 2,2-diphenyl-1,3-benzodioxole has been established so far, other benzodioxoles have been identified as potent biomedical agents. 1,3-Benzodioxole has been reported to have a strong antioxidative capacity [50]. 1,3-Benzodioxole-pyrimidine and its derivatives have been identified as potential succinate dehydrogenase inhibitors and as antifungal agents [51]. The same compound has also been reported to have anticancer properties in both in vitro and in vivo studies against breast, liver, bone, and lung cancers [52].

## 5. Conclusions

Non-small cell lung tumors have multiple genetic abnormalities that can be targeted, and medications that address these variations have been recommended for the treatment of progressed NSCLC subjects. *Moringa oleifera* is one of the most crucial medicinal plants, which demonstrated several medicinal utilities as antitumor, anti-inflammatory, antidiabetic, and hepatoprotective properties. The current study employed various in silico techniques to screen for potential druggable bioactive ligands present in *Moringa oleifera*. 2,2-Diphenyl-1,3-benzodioxole displayed potential pharmacological properties to address the MAGE-A protein, one of the prominent protein(s) in NSCLC tumorigenesis. The ligand 2,2-diphenyl-1,3-benzodioxole actively binds to the branched-chain amino acids of MAGE-A.

## Figures and Tables

**Figure 1 cancers-14-05052-f001:**
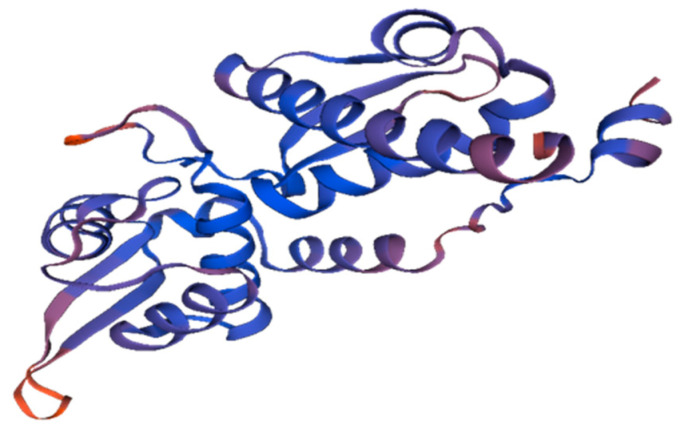
The 3D Structure of MAGE-A build using Swiss Model.

**Figure 2 cancers-14-05052-f002:**
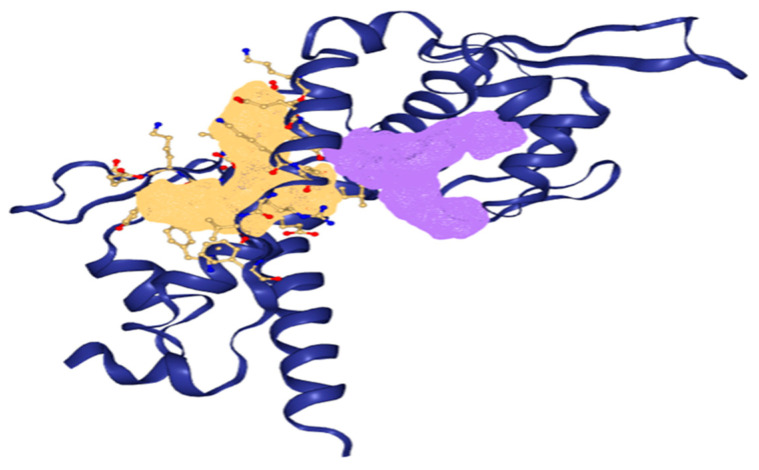
Binding pockets of MAGE-A (the pocket represented in yellow is pocket 1, and the pocket represented in purple is pocket 2).

**Figure 3 cancers-14-05052-f003:**
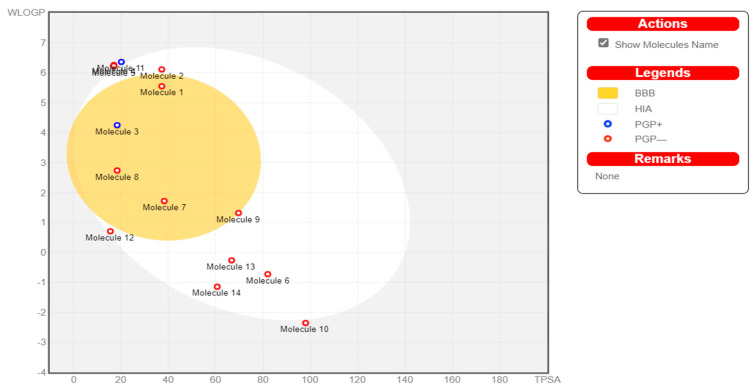
Boiled-Egg assessment: Molecule 1—hexadecanoic acid; Molecule 2—cis-vaccenic acid; Molecule 3—2,2-diphenyl-1,3-benzodioxole; Molecule 4—cyclohexanone,2-(3-chloro-2-butenyl) -2-methyl-6,6-diphenyl; Molecule 5—palmitoyl chloride; Molecule 6—piperazinedone, 4-benzoyl, 2-oxime; Molecule 7—3-chloro-N-isochroman-1-ylmethyl-propionamide; Molecule 8—(Z)-1-(1-Ethoxyethoxy)hex-3-ene; Molecule 9—phenacylidene diacetate; Molecule 10—L-galactose, 6-deoxy; Molecule 11—3,7,11,15-tetramethylhexadec-2-en-1-ol/phytol; Molecule 12—2-pyrazoline,1-isopropyl-5-methyl; Molecule 13—4H-pyran-4-one, 2,3-dihydro-3,5-dihydroxy, 6-methyl; Molecule 14—1,2,3-cyclopentanetriol.

**Figure 4 cancers-14-05052-f004:**
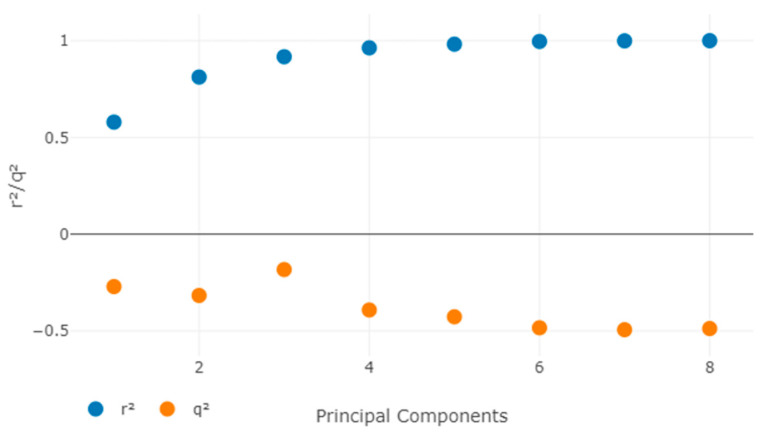
Alignment map of least active compounds.

**Figure 5 cancers-14-05052-f005:**
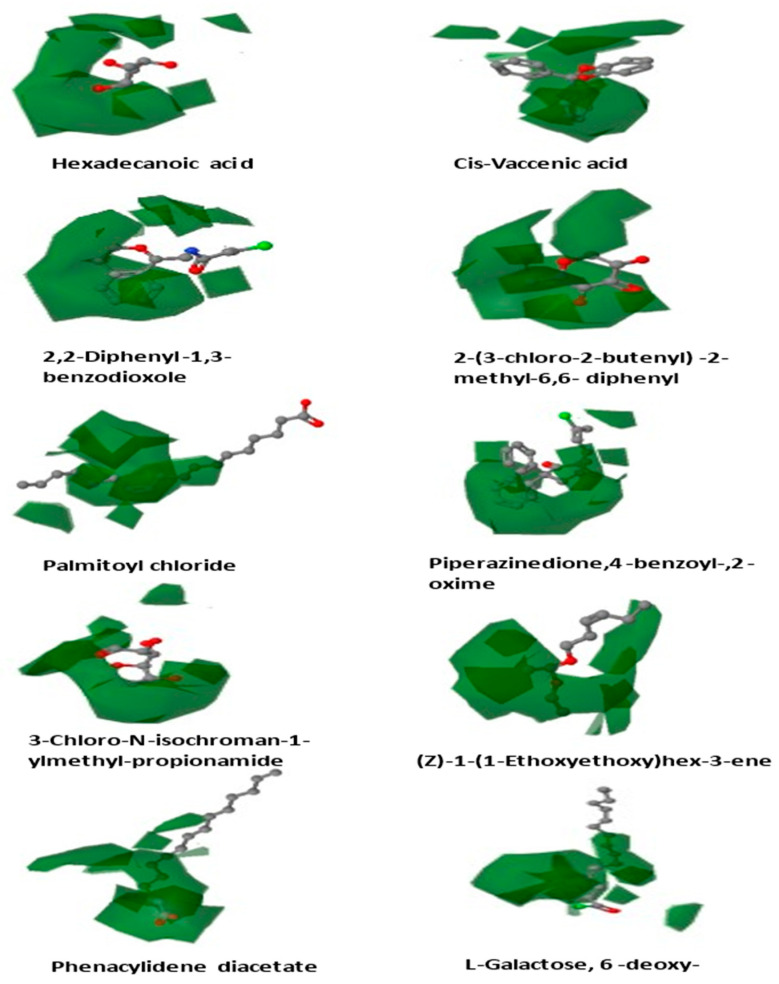
Contour maps of all ligands.

**Figure 6 cancers-14-05052-f006:**
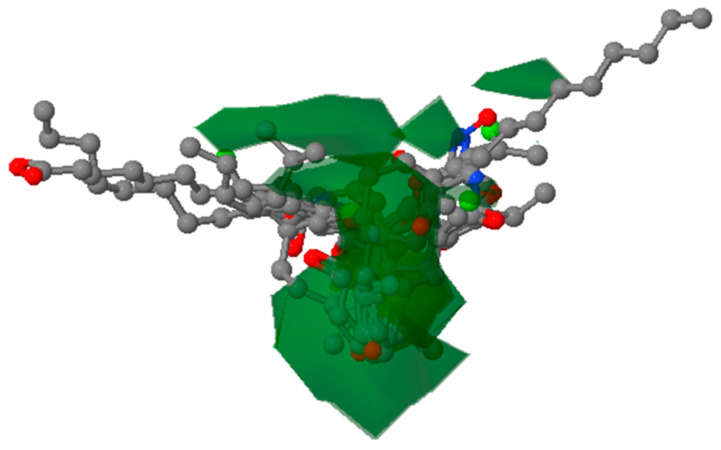
Contour maps of 2,2-diphenyl-1,3-benzodioxole with all conformers.

**Figure 7 cancers-14-05052-f007:**
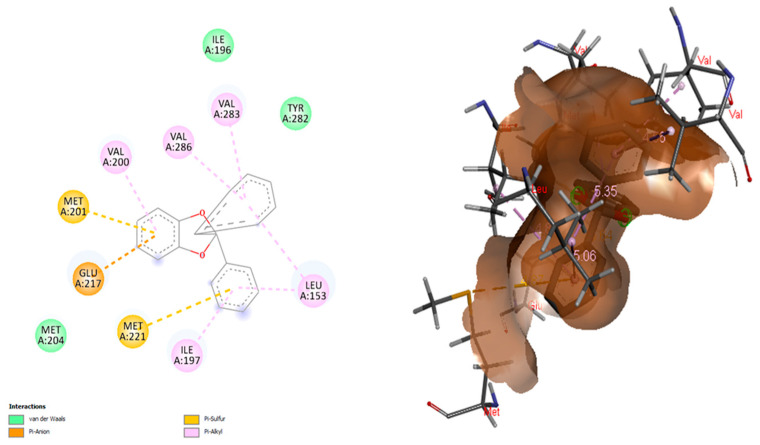
Visualization of molecular interactions of MAGE-A with 2,2-diphenyl-1,3-benzodioxole.

**Figure 8 cancers-14-05052-f008:**
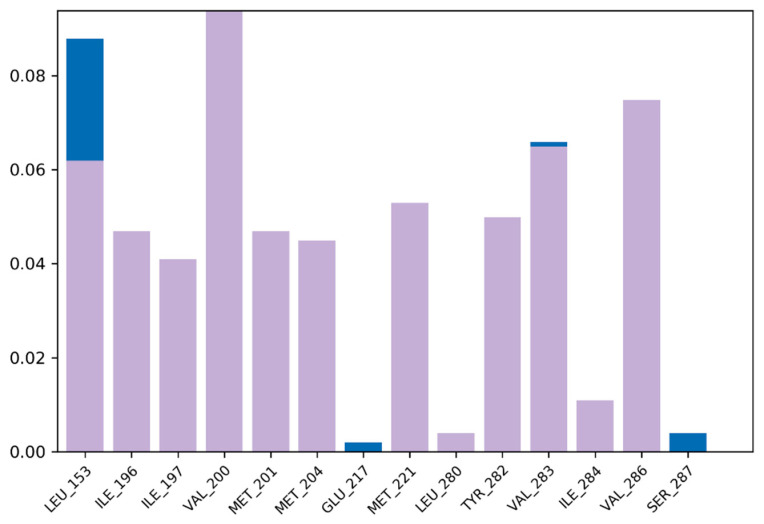
Topmost amino acid trajectories using standard complex.

**Figure 9 cancers-14-05052-f009:**
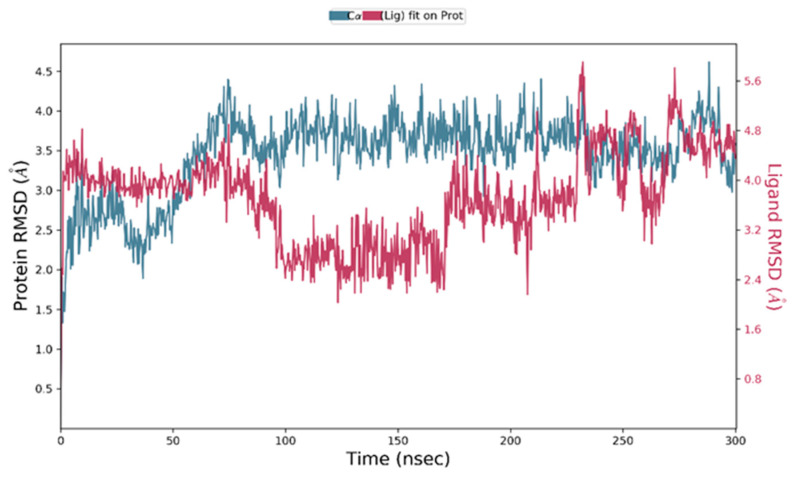
Root mean square deviation of 2,2-diphenyl-1,3-benzodioxole with MAGE-A.

**Figure 10 cancers-14-05052-f010:**
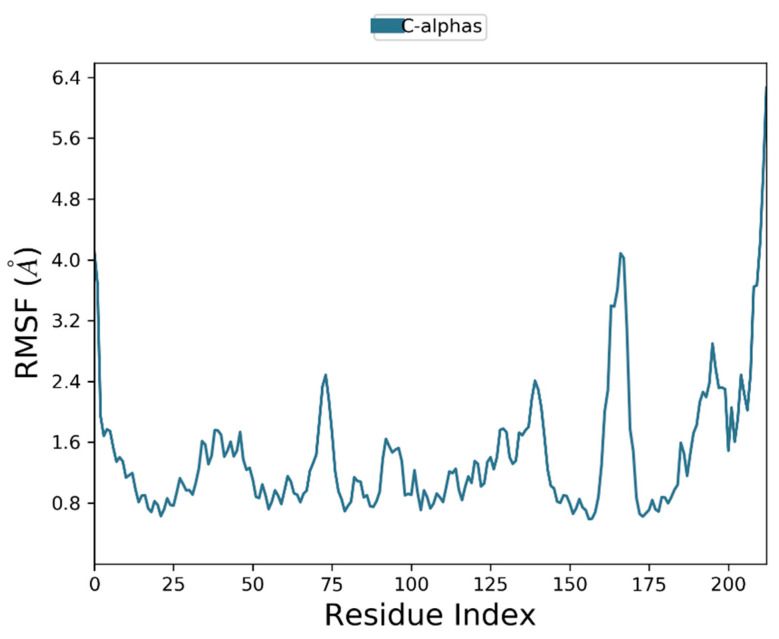
Root mean square fluctuation of MAGE-A.

**Figure 11 cancers-14-05052-f011:**
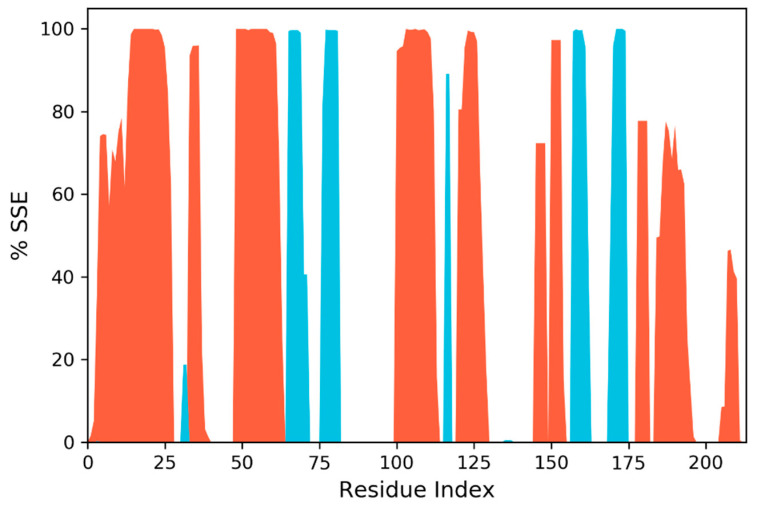
Protein secondary structure histogram.

**Table 1 cancers-14-05052-t001:** Canonical smiles of selected ligands.

Compound No.	Compound	PubChem CID	Canonical Smiles
1	Hexadecanoic acid	985	CCCCCCCCCCCCCCCC(=O)O
2	Cis-Vaccenic acid	5282761	CCCCCCC=CCCCCCCCCCC(=O)O
3	2,2-Diphenyl-1,3-benzodioxole	343959	C1=CC=C(C=C1)C2(OC3=CC=CC=C3O2)C4=CC=CC=C4
4	Cyclohexanone,2-(3-chloro-2-butenyl) -2-methyl-6,6-diphenyl	57116063	CC(=CCC1(CCCC(C1=O)(C2=CC=CC=C2)C3=CC=CC=C3)C)Cl
5	Palmitoyl chloride	8206	CCCCCCCCCCCCCCCC(=O)Cl
6	Piperazinedione,4-benzoyl-,2-oxime	135594388	C1C(=NO)NC(=O)CN1C(=O)C2=CC=CC=C2
7	3-Chloro-N-isochroman-1-ylmethyl-propionamide	583868	C1COC(C2=CC=CC=C21)CNC(=O)CCCl
8	(Z)-1-(1-Ethoxyethoxy)hex-3-ene	108504	CCC=CCCOC(C)OCC
9	Phenacylidene diacetate	569561	CC(=O)OC(C(=O)C1=CC=CC=C1)OC(=O)C
10	L-Galactose, 6-deoxy-2-O-methyl-	169586	CC(C(C(C(C=O)O)O)O)O
11	3,7,11,15-tetramethylhexadec-2-en-1-ol/phytol	145386	CC(C)CCCC(C)CCCC(C)CCCC(=CCO)C
12	2-Pyrazoline,1-isopropyl-5-methyl	573933	CC1CC=NN1C(C)C
13	4H-Pyran-4-one, 2,3-dihydro-3,5-dihydroxy-6-methyl	119838	CC1=C(C(=O)C(CO1)O)O
14	1,2,3-Cyclopentanetriol	92530	C1CC(C(C1O)O)O

**Table 2 cancers-14-05052-t002:** Properties of binding pocket.

Pocket Name	Volume Å3	Surface Å2	Drug Score	H-Bond Acceptors	Hydrophobic Interactions	Pocket Atoms	Apolar Amino Acid Ratio	Polar Amino Acid Ratio
Pocket 1	561.86	872.29	0.77	40	47	124	0.33	0.33
Pocket 2	517.12	764.62	0.77	37	49	125	0.56	0.28

**Table 3 cancers-14-05052-t003:** Physiochemical properties of the ligand molecules.

Compound	Molecular Weight	Heavy Atoms	Aromatic Heavy Atoms	TPSA
Hexadecanoic acid	256.42	18	0	37.3
Cis-Vaccenic acid	282.46	20	0	37.3
2,2-Diphenyl-1,3-benzodioxole	274.31	21	18	18.46
Cyclohexanone,2-(3-chloro-2-butenyl) 2-methyl-6,6-diphenyl	352.9	25	12	17.07
Palmitoyl chloride	274.87	18	0	17.07
Piperazinedione,4-benzoyl-,2-oxime	233.22	17	6	82
3-Chloro-N-isochroman-1-ylmethyl-propionamide	253.72	17	6	38.33
(Z)-1-(1-Ethoxyethoxy)hex-3-ene	172.26	12	0	18.46
Phenacylidene diacetate	236.22	17	6	69.67
L-Galactose, 6-deoxy	164.16	11	0	97.99
3,7,11,15-tetramethylhexadec-2-en-1-ol/phytol	296.53	21	0	20.23
2-Pyrazoline, 1-isopropyl-5-methyl	126.2	9	0	15.6
4H-Pyran-4-one, 2,3-dihydro-3,5-dihydroxy-6-methyl	144.13	10	0	66.76
1,2,3-Cyclopentanetriol	118.13	8	0	60.69

**Table 4 cancers-14-05052-t004:** Evaluation of Lipinski filter.

Compound	Molecular Weight	H-Bond Acceptors	H-Bond Donors	Molar Refractivity	Consensus Log P
Hexadecanoic acid	256.42	2	1	80.8	5.2
Cis-Vaccenic acid	282.46	2	1	89.94	5.68
2,2-Diphenyl-1,3-benzodioxole	274.31	2	0	81.36	4.25
Cyclohexanone,2-(3-chloro-2-butenyl) -2-methyl-6,6-diphenyl	352.9	1	0	106	5.62
Palmitoyl chloride	274.87	1	0	84.02	5.96
Piperazinedione,4-benzoyl-,2-oxime	233.22	4	2	67.09	0.45
3-Chloro-N-isochroman-1-ylmethyl-propionamide	253.72	2	1	67.18	2.06
(Z)-1-(1-Ethoxyethoxy)hex-3-ene	172.26	2	0	51.88	2.6
Phenacylidene diacetate	236.22	5	0	58.43	1.62
L-Galactose, 6-deoxy-	164.16	5	4	35.8	–1.49
3,7,11,15-tetramethylhexadec-2-en-1-ol/phytol	296.53	1	1	98.94	6.22
2-Pyrazoline,1-isopropyl-5-methyl	126.2	1	0	47.47	1.29
4H-Pyran-4-one, 2,3-dihydro-3,5-dihydroxy-6-methyl	144.13	4	2	32.39	–0.22
1,2,3-Cyclopentanetriol	118.13	3	3	27.52	–0.58

**Table 5 cancers-14-05052-t005:** ADME analysis.

Molecule Number	Compound	GI Absorption	BBB Permeability	Pgp Substrate	Silicos-IT LogSw	Silicos-IT Class
1	Hexadecanoic acid	High	Yes	No	–5.31	Moderately soluble
2	Cis-Vaccenic acid	High	No	No	–5.39	Moderately soluble
3	2,2-Diphenyl-1,3-benzodioxole	High	Yes	Yes	–7.17	Poorly soluble
4	Cyclohexanone,2-(3-chloro-2-butenyl) -2-methyl-6,6-diphenyl	Low	No	No	–8.29	Poorly soluble
5	Palmitoyl chloride	Low	No	No	–6.5	Poorly soluble
6	Piperazinedione,4-benzoyl-,2-oxime	High	No	No	–2.2	Soluble
7	3-Chloro-N-isochroman-1-ylmethyl-propionamide	High	Yes	No	–4.57	Moderately soluble
8	(Z)-1-(1-Ethoxyethoxy)hex-3-ene	High	Yes	No	–2.27	Soluble
9	Phenacylidene diacetate	High	Yes	No	–2.56	Soluble
10	L-Galactose, 6-deoxy-	Low	No	No	1.9	Soluble
11	3,7,11,15-tetramethylhexadec-2-en-1-ol/phytol	Low	No	Yes	–5.51	Moderately soluble
12	2-Pyrazoline,1-isopropyl-5-methyl	High	No	No	–0.8	Soluble
13	4H-Pyran-4-one, 2,3-dihydro-3,5-dihydroxy-6-methyl	High	No	No	0.15	Soluble
14	1,2,3-Cyclopentanetriol	High	No	No	1.2	Soluble

**Table 6 cancers-14-05052-t006:** Toxicity prediction.

Compound	Toxic Dose (LD_50_ Values in mg/kg Body Weight)	Toxicity Class	Hepatotoxicity
Hexadecanoic acid	900	Class IV	No
Cis-Vaccenic acid	48	Class II	No
2,2-Diphenyl-1,3-benzodioxole	720	Class IV	No
Cyclohexanone,2-(3-chloro-2-butenyl) -2-methyl-6,6-diphenyl	750	Class IV	No
Palmitoyl chloride	400	Class IV	No
Piperazinedione,4-benzoyl-,2-oxime	6800	Class VI	No
3-Chloro-N-isochroman-1-ylmethyl-propionamide	380	Class IV	No
(Z)-1-(1-Ethoxyethoxy)hex-3-ene	5000	Class V	No
Phenacylidene diacetate	5000	Class V	No
L-Galactose, 6-deoxy	23,000	Class VI	No
3,7,11,15-tetramethylhexadec-2-en-1-ol/phytol	5000	Class V	No
2-Pyrazoline,1-isopropyl-5-methyl	800	Class IV	No
4H-Pyran-4-one, 2,3-dihydro-3,5-dihydroxy-6-methyl	595	Class IV	No
1,2,3-Cyclopentanetriol	12,500	Class VI	No

**Table 7 cancers-14-05052-t007:** Biological activity and conformer of compounds.

Property	GM Label	GM ID	GM Conf	Longest Label	Longest ID	Longest Conf
Least Active	2,2-Diphenyl-1,3-benzodioxole	3623329	0	2,2-Diphenyl-1,3-benzodioxole	3623329	2
Most Active	Cyclohexanone,2-(3-chloro-2-butenyl) -2-methyl-6,6-diphenyl	3623333	0	Cyclohexanone,2-(3-chloro-2-butenyl) -2-methyl-6,6-diphenyl	3623333	6
Heaviest	Cyclohexanone,2-(3-chloro-2-butenyl) -2-methyl-6,6-diphenyl	3623333	0	Cyclohexanone,2-(3-chloro-2-butenyl) -2-methyl-6,6-diphenyl	3623333	6
Longest	Palmitoyl chloride	3623337	0	Cis-Vaccenic acid	3623332	12
Most Flexible	Phytol	3623339	0	Phytol	3623339	12
Most Rigid	2,2-Diphenyl-1,3-benzodioxole	3623329	0	2,2-Diphenyl-1,3-benzodioxole	3623329	2
Least Polar	Pyrazoline	3623341	0	Pyrazoline	3623341	0
Most Polar	Fucose	3623334	0	Fucose	3623334	0

**Table 8 cancers-14-05052-t008:** The results of r^2^, q^2^, and the corresponding SDEC and SDEP values.

PC	r2	SDEC	q2	SDEP
1	0.579	0.203	–0.271	0.354
2	0.812	0.136	–0.317	0.36
3	0.917	0.091	–0.183	0.341
4	0.963	0.06	–0.392	0.37
5	0.982	0.042	–0.427	0.375
6	0.996	0.02	–0.484	0.382
7	0.999	0.009	–0.494	0.383
8	1	0.005	–0.488	0.383

**Table 9 cancers-14-05052-t009:** Experimental values for the selected compounds.

Sl No	Name	Experimental Activity
1	1,2,3-Cyclopentanetriol	7.63827
2	2,2-Diphenyl-1,3-benzodioxole	7.63827
3	3-Chloro-N-isochroman-1-ylmethyl-propionamide	7.33724
4	4H-Pyran-4-one, 2,3-dihydro-3,5-dihydroxy-6-methyl	8.22185
5	Cis-Vaccenic acid	8.22185
6	Cyclohexanone,2-(3-chloro-2-butenyl) -2-methyl-6,6-diphenyl	7.92082
7	L-Galactose, 6-deoxy-	8.22185
8	(Z)-1-(1-Ethoxyethoxy)hex-3-ene	8.22185
9	hexadecanoic acid	8.22185
10	Palmitoyl chloride	7.49485

**Table 10 cancers-14-05052-t010:** Docking score of MAGE-A protein with selected ligands.

Compound	Binding Affinity
3-Chloro-N-isochroman-1-ylmethyl-propionamide	–5.5
(Z)-1-(1-Ethoxyethoxy)hex-3-ene	–3.3
4H-Pyran-4-one, 2,3-dihydro-3,5-dihydroxy-6-methyl	–3.8
Cis-Vaccenic acid	–4
2-(3-chloro-2-butenyl) -2-methyl-6,6-diphenyl	–6.3
1,2,3-Cyclopentanetriol	–3.6
2,2-Diphenyl-1,3-benzodioxole	–7.3
L-Galactose, 6-deoxy-	–3.4
Hexadecanoic acid	–4.1
Palmitoyl chloride	–4.1
Phenacylidene diacetate	–4.9
3,7,11,15-tetramethylhexadec-2-en-1-ol	–4.3
2-Pyrazoline,1-isopropyl-5-methyl	–3.6
Piperazinedione,4-benzoyl-,2-oxime	–4.4

## Data Availability

The authors hereby confirm that all the data of this research are available within this manuscript.

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
