# Peer review of "Virtual Screening and Quantitative Structure–Activity Relationship of Moringa oleifera with Melanoma Antigen A (MAGE-A) Genes against the Therapeutics of Non-Small Cell Lung Cancers (NSCLCs)"

_cancers, 2022, doi:10.3390/cancers14205052_

Round 1
Reviewer 1 Report
Good quality paper.
Author Response
We thank the reviewer for the prompt comments on our article. We have made the necessary changes in the article as suggested by the reviewer.
Reviewer 2 Report
In this manuscript, the authors screened 14 phytocompounds from Moringa oleifera and found that the binding of 2,2-Diphenyl-1,3-benzodioxole and MAGE-A yielded the highest binding energy. Thus, 2,2-Diphenyl-1,3-benzodioxole is a candidate drug for targeting MAGEA in NSCLC. The manuscript looks good, but the writing needs to be polished. There are also a couple of concerns that need to be addressed.
1. To validate the binding between 2,2-Diphenyl-1,3-benzodioxole and MAGEA, DARTS and CETSA should be conducted.
2. The toxicity of 2,2-Diphenyl-1,3-benzodioxole on NSCLC cell lines should be determined. This data would be more convincing for your conclusion.
Reviewer 3 Report
In this article, K Prasad et al. investigate the bioactive phytochemicals found in Moringa oleifera in order to evaluate their potential role to prevent MAGE-A engagment in NSCLC. They identified the 2,2-Diphenyl-1,3-benzodioxole was stipulated as lead candidate with potential pharmacological properties to address MAGE-A proteins counteracting NSCLC tumorigenesis.
The article is very interesting and provides important information for the development of new immunotherapeutic drugs for the treatment of NSCLC.
I strongly recommend the authors to evaluate in the future the in vitro anti-tumor and immunomodulatory effects of the lead candidate.
Author Response
Thanking the reviewer for suggesting an investigation of immunomodulatory effects, we appreciate the positive comments. We intend to do the needful evaluation in the future course of this study.
Round 2
Reviewer 2 Report
The Authors have addressed all of my concerns with the original manuscript. The revised manuscript is ready for publication.